# A Framework for Selecting and Assessing Wearable Sensors Deployed in Safety Critical Scenarios

**DOI:** 10.3390/s24144589

**Published:** 2024-07-15

**Authors:** Robert Houghton, Alberto Martinetti, Arnab Majumdar

**Affiliations:** 1Centre for Transport Studies, Imperial College London, London SW7 2AZ, UK; 2Design, Production and Management Department, University of Twente, 7522 NB Enschede, The Netherlands

**Keywords:** wearable sensors, psychophysiology, usability, safety critical, design

## Abstract

Wearable sensors for psychophysiological monitoring are becoming increasingly mainstream in safety critical contexts. They offer a novel solution to capturing sub-optimal states and can help identify when workers in safety critical environments are suffering from states such as fatigue and stress. However, sensors can differ widely in their application, design, usability, and measurement and there is a lack of guidance on what should be prioritized or considered when selecting a sensor. The paper aims to highlight which concepts are important when creating or selecting a device regarding the optimization of both measurement and usability. Additionally, the paper discusses how design choices can enhance both the usability and measurement capabilities of wearable sensors. The hopes are that this paper will provide researchers and practitioners in human factors and related fields with a framework to help guide them in building and selecting wearable sensors that are well suited for deployment in safety critical contexts.

## 1. Introduction

Increasingly, the usage of wearable sensors is becoming commonplace both in everyday life and for research. The rise of small, powerful and low-cost computing has brought physiological sensing to the masses. Additionally, many researchers and companies now rely upon wearable sensors to collect data to explore psychophysiological phenomena, such as stress, fatigue and cognitive workload in safety critical contexts. An advantage wearable sensors confer is that they enable large datasets to be collected due to their sampling frequency and generally long battery life, and indeed they contribute to a major revolution in big data revolution. This in turn enables previously unconsidered research questions to be explored, which technical limitations on data collection in the past had rendered impossible to study.

All the indications are that the number of wearables being developed and entering the market is going to increase exponentially in the foreseeable future. Given this, there is hence a need for a framework to assess these devices, especially those used in operational and safety critical contexts such as aviation, rail, road, defense, construction, and energy. Compounding this is that many researchers in the wearable space domain are focused on machine learning, often reducing the role of the sensor as a means of data collection. For such researchers, their sole concern is the model developed from the data, yet this ignores how their model will perform under differing conditions if the sensor is deployed in operational contexts, i.e., rugged conditions where movement, sweat, humidity, sand/dust and vibration may all impact data quality [1]. Whilst context may alter the parameters needed, e.g., working in air traffic control will likely result in a different sensor compared to construction, to ignore the usability of the sensor seems egregious. However, there is yet to be a single unified source of understanding of the usability factors related to the successful adoption of sensors. Thus, there is a need for guidance, such that researchers and organisations can select appropriate sensors that also are theoretically acceptable.

A framework would be useful not only in assessing current devices on the market but also guide designers and developers to create devices which maximize the principles of measurement and usability, as well as provide design guidance. Currently, no framework exists which is sufficiently comprehensive to evaluate current technology in its entirety as well as to provide guidance in developing future devices. Hence this paper attempts to outline a framework which is comprehensive enough to allow for a good level of evaluation. In order to do this, the framework attempts to review parameters that are relevant to success, i.e., usability and acceptability. At the outset, it is important to recognize that the purpose of this paper is not to provide a systematic literature review, but rather to provide a succinct summary of the different parameters needed to develop and assess wearable sensors for psychophysiological monitoring in safety critical contexts.

### 1.1. Theoretical Background

#### 1.1.1. Foundations of Wearables—Measurement and Usability

Wearable sensors offer both a convenient means and a novel methodology for measuring psychophysiological phenomena, which until recently was not attainable. Constructs such as stress fatigue and cognitive workload have a history of being detected through psychophysiology [2,3]. Recently, researchers have managed to detect these phenomena outside of the laboratory, such as stress in construction workers [4], fatigue in bus drivers [5], and cognitive workload in surgeons [6]. However, for these devices to be accepted and used in operational contexts, they need to be a valid and reliable measurement tool that can capture and measure the required phenomena during operations. In addition, the device needs to be user-friendly, promoting an enjoyable and purposeful user experience [7]. To assess this, there is a need to consider differing aspects of the efficacy of the device as a measurement tool across several different parameters, such as validity, both construct and predictive reliability, sensitivity, selectivity, and generalizability [8]. Furthermore, usability can be assessed by considering the ease of interface with the device, its robustness in the presence of water and environmental extremes, as well as the size, weight and fit of the device [9]. Assessing usability can be thought of with the question, “Can I still complete my job whilst wearing this device and does the device not intrude on my job?”, e.g., the device first of all does not prevent the operator from completing their job and, secondly, it does not add a negative stimulus (in the form of a wearable sensor) that makes their job harder. For example, certain wearable ECGs may be unsuitable for roofing work due to impediments to the flexion of the human body.

Dinges and Mallis’s (1998) [10] seminal paper on the physiological monitoring of fatigue outlines several measurement parameters, such as validity, reliability, sensitivity, and specificity which must be considered for devices, such as EEGs, ECGs and eye trackers, before they can be considered a tool for measuring fatigue. While these parameters were classified as scientific/engineering criteria by the authors in 1998, much of what they discussed is relevant, not just for fatigue monitoring, but also for monitoring any psychophysiological state in and outside of the laboratory environment. However, these considerations or criteria as listed by [10] are of even greater importance when such devices are intended for use in a safety critical context.

For example, to use devices that capture data in real-time for a machine learning model so as to create an output would require that device to capture data that is valid and reliable for a length of time, e.g., using a wearable ECG to capture real-time HRV data, which is then filtered and analyzed in real-time to capture fatigue in long haul truck drivers. If operators and organizations are reliant on devices to provide an additional safety net, if these are not carefully selected and benchmarked for the specific use [11] then the device is unsuitable for use. Thus, optimizing wearable sensors as a measurement tool is one of the key tenets when designing or evaluating a sensor for both research and safety applications.

Ref. [10] also briefly consider usability in what they term “implementation criteria”: how will the device be used, is the device unobtrusive, and what will it do, e.g., type of alarm? While this is commendable, understanding the usability of devices actually must go beyond this in order to be meaningful to the user in achieving a goal or outcome [7].

To this end, ref. [12] suggest that researchers and designers need to understand why people may engage or disengage with sensors in clinical applications. This sentiment, though, can also be extended to other safety critical environments. For example, if a set of mine workers are asked to wear a fatigue monitoring device because they work night shifts and drive several hours to and from the mine site before and after their 8–10-h shift, and many of them are reporting that tiredness is affecting their concentration, the hope is that they would wear the device to obtain meaningful data.

However, if the device is complex to use, fits poorly and is uncomfortable, the workers may stop wearing the device altogether [13]. Furthermore, the device should not just benefit those interested, but the data and outputs should be formatted in such a way that they can be used by researchers, designers and non-research users. For example, researchers may be interested in signal quality, whereas the mine workers may want a simple output telling them about different gradations of fatigue to make more positive changes to their sleep schedule. Understanding why devices may elicit a negative or positive user experience is important for long-term use and engagement.

#### 1.1.2. Using Design to Augment Measurement and Usability

To truly create a device that scores highly on measurement and usability, there is also a need to consider the design of the device. If measurement and usability are the two pillars to creating a device that will excel in the field, then design is the mortar that holds the pillars together and enriches both measurement and usability. Consider the below example.

A Holter monitor is worn on a belt or a lanyard around the neck, with electrodes physically adhering to the chest for a certain length of time. This is a clinically validated ECG, which is suitable for monitoring patients but is unsuitable for monitoring psychophysiological functioning amongst construction site workers. The Polar H10 is a commonly used ECG device and is simply a combined sensing unit and set of electrodes embedded in a simple chest strap. This is most likely to be a better choice for construction workers because it is discrete and lacks any external wires, maximizing comfort without the risk of the wires catching on any objects. Here, the physical design of the device, e.g., bulkiness, external wires, electrode type and placement, infer that possible use cases for the devices are optimized. This is an example of how design can enhance usability. The Polar H10 began as an activity monitor before it saw a decent uptake from the academic community as a good ECG to use to monitor HRV [14]. Many of the design choices made when developing the H10, such as simplicity of use, low cost, and discreet nature, make it an excellent choice for measuring HRV in both laboratory and field research. The polar H10 refers to the unit itself, which is often paired with a Polar Pro strap to create the ECG monitoring device. It switches on automatically after wetting the electrodes and being placed around the chest to detect a heartbeat. The electrodes are embedded within the strap and make contact with the body through a rubber silicone material. The absence of metal or conductive gel that sticks to the body greatly reduces the chances of skin irritation. The flexibility of the strap means that the H10 device can be fitted well and comfortably on the body, meaning it can be worn without creating excessive interference in movements. Here, the design can be seen as fulfilling two functions: (i) maximizing good measurement by having a flexible and customizable electrode that will curve with the body keep contact over the chest muscles and flex with respiration, (ii) by using a simple chest band and an automated start on the device, the user can simply plug and play the device, obtaining instant heart rate data, leading to high usability levels [14].

## 2. Methodology and Paper Selection

To highlight relevant parameters related to wearable sensors, papers were identified that had previously outlined parameters important for psychophysiological monitoring. From this initial selection, a general list of parameters was generated. Whilst these papers highlighted parameters, not all parameters were necessarily concerned with the sensor itself, but rather the management and integration of sensors once they are deployed. Management and integration are important considerations, and users and researchers interested in wearable sensors should not ignore the practicalities of what will occur once a sensor has been deployed. However, such considerations are beyond the scope of this paper, so any parameters deemed not to be core to the sensor as a sensing entity were removed. Additionally, sometimes a parameter, e.g., toughness, was present in more than one paper, so any repetitions were removed. Additionally, whilst papers that concern aspects such as machine learning and model validation are interesting and related to wearable sensors, they are slightly removed from the primary sensor eco-system, unless a sensor has onboard processing capabilities. There is also extant literature and reviews on machine learning and wearable sensors for safety critical contexts, such as [15,16].

### 2.1. Steps in Paper Identification

Since holistic papers that discuss the parameters that wearable sensors need for safety critical contexts are rare, ref. [10] were used to generate the initial source list of parameters. Ref. [10] list 16 parameters related to scientific, practical and legal criteria. They consider both the sensor itself and other aspects, such as ethics and legality. They are also constrained to fatigue measurement. The authors mainly took scientific and practical criteria, reclassifying them through discussions into either measurement or usability. We then searched for papers related to these constructs, such as what [10] call Robustness, or toughness, with water tolerance being somewhat adjacent to toughness. This process was carried out in an organic manner. See Figure 1 for a flow chart of paper selection.

After searching for papers concerning all these aspects of wearable sensor selection and development, a list of 22 key parameters was identified. Table 1 outlines the parameters identified together with their source material. All parameters have been chosen due to their contribution to the selection and use of wearable sensors in safety critical contexts. These parameters are related directly to psychophysiological sensors, and most will also extrapolate to wearable sensors in general, with only electrode type and diagnosticity unique to psychophysiological wearable sensors. The list is not exhaustive but aims to bring together most common fields related to wearables from some aspects, including material science, electronics and electrical engineering, human factors and ergonomics, user centered design, and computing.

### 2.2. Parameter Classification

Once the parameters were identified, they were subsequently classified based upon a measurement, usability or design principle. The basis for this was how the parameters related to the sensor. Whilst the type of electrode can influence measurement, this is a design choice of the device, in contrast to validity. Users cannot pick and choose the types of validity for their device, rather this is a property that exists inherently to measurement science. In general, usability relates to parameters that do not concern measurement but are neither specifically design choices, such as the degree of wearability, nor concern individual differences that will impact the overall user experience of the device. Design refers to the choices manufacturers make that can impact either usability or measurement, such as the sampling rate or water tolerance. Section 3.1, Section 3.2 and Section 3.3 of the paper discuss the parameters of measurement, usability and design and why they are important in device selection and assessment in safety critical contexts.

## 3. Results

This section describes each parameter, and its importance to the assessment and deployment of wearable sensors for psychophysiological monitoring in the field. Each parameter is listed under its classification, resulting in three sections: measurement, usability and design.

### 3.1. Measurement Principles

When states such as fatigue, stress, and cognitive workload are measured, the hope is that the device will both accurately measure what it attempts to measure, and that it will be able to measure these states repeatedly. To assess the accuracy and consistency of measurement, there are several scientific principles that need to be considered when evaluating wearable devices.

#### 3.1.1. Sensitivity

At its core, sensitivity can be summed up with the question, “Does the device detect the event we are trying to measure?” For example, can a device detect fatigue or no fatigue? Another way to conceptualize this idea is to consider whether the device will miss a fatigue event, and if this happens then the device can be said to lack sensitivity [27]. Furthermore, sensitivity has also been used to assess whether a device is sensitive to increases or decreases in the phenomena the device is trying to measure. If a measure is sensitive to mental workload, task demands would be expected to increase both the number of items remembered, and the number of calculations required, or else add multiple time pressures [28]. For example, ref. [29] found skin conductance and heart rate increased linearly with increases in driving difficulty, from low, medium, to high difficulties. Moreover, ref. [30] found neuro-imaging sensors, such as ECG and fNIRS, to be superior to cardiac and ocular sensors in detecting differences in workload across several wearable sensors. Hence, devices need to demonstrate how to assess whether psychophysiological measures are sensitive to increases in workload, with increases in task demands. Similar practices can be used for stress and fatigue, with low, medium, and high stress or fatigue conditions. Ref. [27] suggest that, in system evaluations, very fine granularity with respect to sensitivity is very useful, as even small changes in interfaces can lead to disastrous outcomes.

#### 3.1.2. Specificity

Specificity, whilst having some overlap with reliability, is concerned with the consistency of correct measurement. Will a device accurately identify an individual as not fatigued when they are indeed not fatigued? Hence, this relates to how often the device elicits a false alarm within a proportion of operators or throughout an operator’s shift [7]. Specificity is important as too many false alarms can reduce trust in the device and thereby cause delays and disrupt working patterns (https://www.ukconstructionmedia.co.uk/case-study/smartcap-validation-independent-assessment-from-universidad-de-chile/2, accessed on 14 October 2018). Having more liberal specificity levels can also have dire consequences. An example of this can be seen from April 2016 when three fishermen drowned off the coast of Scotland due to flooding in the interior of their trawler. Whilst the vessel had alarms to alert the crew members to any flooding incident, excessive false alarms waking the crew during sleep led them to deactivate the alarm (https://www.gov.uk/maib-reports/sinking-of-vivier-creel-boat-louisa-with-loss-of-3-lives, accessed on 5 July 2024).

Specificity and sensitivity have a proportional relationship so, as one increases, the other decreases and hence it is important to strike the correct balance [31]. Ideally, a device should capture enough events stemming from the construct being measured, as well as the level of the construct, whether it be workload, fatigue, or stress, in addition to being sufficiently conservative regarding false alarms so that the operator’s work is not disrupted, whilst preserving trust in the device.

#### 3.1.3. Validity

Validity refers to the accurate measurement of phenomena, often referred to as construct validity in the measurement literature and plays an important role in psychophysiological measurements [32]. A device or measure is said to have good construct validity if it is found to measure the actual phenomenon it claims to measure, such as fatigue, rather than stress. Traditionally the assessment of construct validity would assess the level of convergent and discriminant validity a measure or sensor has compared to similar or related measures, such as comparing a wearable ECG to a traditional clinical ECG [17].

Convergent validity is the degree of correlation between measures of the same construct. For example, if a workload questionnaire correlates highly with other prior well-validated measures of workload, such as performance measures, self-reports, or psychophysiological measures, it can be said to have good convergent validity. Discriminant validity, also termed selectivity by [33], is the notion that a measure will only assess the construct it claims to measure and no others. Ref. [33] quote, “The index must be sensitive only to differences in cognitive demands, not to changes in other variables such as physical workload or emotional stress, not necessarily associated with mental workload” (p. 63). For a measure to have discriminant validity, it should be relatively independent of other constructs and not correlate too highly with measures of differing constructs. Whilst this is relatively simple to carry out in self-report measures and scale validation, following the guidelines outlined by [17] this is not so with psychophysiological measures.

Ref. [34] suggest that the assessment of wearable validity can take the form of comparing the wearable to a current gold standard or referent device and outline three approaches. Firstly, they note correlation approaches, such as leveraging correlations to assess agreement between devices or signals. The second approach concerns statistics that describe various error rates, such as mean absolute error (MAE) and mean squared error (MSE). Finally, they describe using visual approaches such as Bland–Altman plots to visually assess signal agreement and overlap. Ref. [34] further note that all validity approaches have set limits as to what indicates validity, with scholars discordant as to what are the best limits, e.g., should correlational approaches be >0.8 or 0.9? Additionally, correlation approaches are criticized for suggesting that high correlations do not necessarily indicate agreement, and visual approaches are open to subjectivity in interpretation to some degree. It may be better to apply an approach at all levels as outlined by [35].

Ref. [35] suggest that validity should be assessed on signal, parameter and event levels. The signal level is the most like-for-like comparison and essentially assesses whether two or more devices generate the same volume of raw data. Parameter level refers to whether devices produce similar parameters, such as HR and SCL. Finally, event level refers to both devices registering the same sensitivity to an event, such as a stimulus or similar change, over time in response to fatigue or stress (See Section 3.1.1 Sensitivity). Ref. [35] aim to develop a standardized test procedure and, whilst this cannot be described in detail in this paper, it offers a useful method to assess the validity of devices, with signal, parameter and event all critical to the real time assessment of psychophysiology in safety critical environments. The authors would argue that event is the most important factor in this method; however, if one device detects fatigue better than another, independent of signal overall or parameter estimation, then surely that is the most appropriate.

#### 3.1.4. Reliability

Reliability is often paired with validity and refers to whether a device or measure is consistent in its measurement [10]. Whilst reliability has been a central theme of research for many years, it is especially important regarding human factors, as measures which are unreliable can have dangerous consequences. For example, an unreliable measure of fatigue or workload could lead operators to work in sub-optimal conditions which then compromises their attention, communication and decision making. One common way to assess the reliability of a device is test–retest reliability, in which multiple trials of a device are conducted to assess the level of correlation between trials.

Ref. [36] notes that, whilst test–retest correlation is a good measure of reliability, care must be paid when comparing a small sub-group of a larger sample. Ref. [36] states that the test–retest correlation is sensitive to the heterogeneity of the data gathered. For example, when looking at a small sub-set, the data may appear to have no correlation at all; however, as the sample size increases, so too does the linearity, increasing the strength of the correlation between the trials. Ref. [35] suggests using the standard error of measurement or typical error, which is the within subject standard deviation, a statistic which captures how much variation one subject displays between trials, e.g., the standard deviation between scores from a fatigue monitoring device across two working weeks. Ref. [35] states that the typical error is the most important reliability measure for researchers as it “affects the precision of estimates of change in the variable of experimental studies” (p. 2). There are also other measures of reliability, such as the limit of agreement, which is represented by a range in which 95% of the individuals scores will lie. The problem with limits of agreement, however, is that the range is dependent on the degrees of freedom and sample size. Hence, a study using eight participants only has seven degrees of freedom, which leads to limits of agreement at 79%, rather than 95%. This, however, is not an issue of typical error, as typical error calculations are independent of the sample size.

Other issues of reliability may stem from the nature of the devices or the environment itself. First, many wearable devices measure electrical signals stemming from the brain or muscles. Traditionally lab-based devices have been notoriously affected by electrical artifacts, which mask the pattern of electrical activity being investigated. This may be compounded by the nature of work using wearable devices, e.g., the life-band devices developed by Smartcap are susceptible to movement artifacts by jaw movements from chewing gum. Similarly [37] note similarities with electro-ocular grams, which are susceptible to extraneous head and eye movement artifacts. These issues can be overcome ideally through design, such as in-ear EEG being far more resistant to movement artifacts than scalp-based measures, or by the removal of electro-signals all together. Additionally, researchers can rely on classification algorithms to extract, clean, and classify data adequately. Furthermore, differing or extreme environments may affect the reliability of measurement. For example, the percentage of eye closure (PERCLOS) measures, whilst providing some degree of fatigue measurement, are not always reliable due to data missing at several points due to either the driver’s eyes moving out of the required field of view to perform checks on accelerometer or mirrors, or ambient light conditions becoming too bright or too low for the camera to accurately assess the percentage of eye closure [38].

These issues can be combated through taking a broader approach in the design phase. For example, ref. [39] developed a wearable eye-based sensor to detect fatigue based on blink information rather than percentage eye closure, which was reliable both in- and outdoors, as well under differing light conditions throughout the day. Ref. [39] further note that the device was designed and evaluated in terms of device shifting round the eyes and user mobility, as these are often not considered in fatigue detection research. Thus, developers and researchers should consider whether a device will also be reliable in ambulatory settings, which may cause movements in the device. Whilst wearable eye-based devices have been used extensively in transport research [39], there may be other contexts, such as search and rescue, enclosed firefighting, mining, and construction and maintenance, in which they can provide in-depth and granular detail. In these scenarios, whether an eye-based device is in a perfect condition for measurement will not be (correctly) at the forefront of the operator’s mind. Thus, future devices, if not being developed specifically for stationary contexts, should consider ways to preserve reliability in ambulatory contexts. There is, however, yet to be any research assessing eye-based wearables in more extreme occupations.

In conclusion, it is paramount that reliability is assessed accurately, as without a high level of consistent measurement, wearable devices are unlikely to provide useful information, and instead cause more stress and loss of time in ascertaining the accuracy of data collected. Furthermore, developers and researchers should consider how the reliability of devices can be preserved across a multitude of different contexts.

#### 3.1.5. Diagnosticity

According to [18] and subsequently [33] and [8] diagnosticity is a unique metric which applies to mental workload assessment. Diagnosticity refers to a device’s ability to detect changes in workload and the reasons for the changes. Ref. [28] suggest that diagnosticity refers to whether a device can differentiate between workload sub-sets, such as the degree of perceptual spatial and psychomotor demands impacting psychological resources. They further argue that diagnosticity is of special importance due to the need to combat performance decrements. If a device can offer insights into what part of the task, environment or organizational contexts is causing sub-optimal performance, then effective interventions can be put in place. This is certainly a necessary requirement for measures of mental workload and both developers and researchers should strive to maximize diagnosticity as much as possible. Whilst diagnosticity does not appear to have been discussed in relation to other cognitive states or phenomena, it is possible that the principle could be useful in the design and development of future sensors, though further research is needed to test this assumption. For example, there is no doubt that a measure of fatigue based on drowsiness would suggest that a lack of sleep is the cause of fatigue. However, this device fails to highlight the aspects of an operator’s lifestyle and job that cause drowsiness, whereas a measure of mental workload can highlight the types of workload being strained in the operator, e.g., time constraints, ambiguity, perceptual overload, and maximum capacity of working memory.

#### 3.1.6. Generalizability

Ref. [10] briefly mention the importance of generalizability, i.e., the notion that a device will measure the same event in everyone. Generalizability can be framed conceptually as well as operationally. Conceptual generalizability is whether the device will measure the construct of fatigue, e.g., hypo-vigilance (lack of attention), identically in all individuals. Operational generalizability, on the other hand, is whether a device can capture fatigue through criteria, such as eye blinks, identically in all individuals. Whilst this is the ideal, ref. [40] notes that it is often difficult to achieve ideal levels of generalizability due to individual differences. For example, PPG performance drops as skin darkens [41] and is impacted by damaged or sensitive skin, such as in those with burns, scars, eczema and the elderly [42]. Here, the device’s data could potentially be compromised because the designer has not considered all use cases or conducted rigorous user testing on various populations who may use the device. Individuals working in safety critical environments are more at risk of damaging the skin, thus PPG devices may need to be highly accurate in order to work across all contexts. Additionally, ref. [43] note that designers should also consider the users’ skin characteristics, such as scars, eczema, rashes and irritation, in relation to comfort and wearability. Therefore, when selecting devices, it is important to be consider whether the device will apply to the full spectrum of the user population.

### 3.2. Usability

While it is of paramount importance that devices are evaluated in accordance with the scientific principles previously mentioned, an extremely strong device will lack usability if it cannot be transferred from a test environment to the field. This section of the framework will therefore discuss the usability features devices should adhere to.

#### 3.2.1. Intrusiveness

Any wearable device should not intrude on the nature of the task [8], i.e., it should not increase task demands or be a cause of distraction for the operator. Devices developed to measure human cognition should always be designed to passively capture the current state of the operator, with little if any input from the operator, as this input increases the demands of the job. Furthermore, wearable sensors should be as transparent as possible, e.g., integrated into hard hats, as well as designed in accordance with human physiology, so that devices do not restrict movement [22]. Ref. [44] note that, even in laboratory studies, some wearable sensors can be intrusive, causing discomfort and distraction due to poor user experience. This is likely to be compounded in working environments, such as in oil and gas, construction, transport maintenance, forestry and fire fighting, all sectors which deviate considerably from laboratory or test conditions. For example, the user experience of wearing a device should be optimized for possible extreme environments. Moreover, ref. [45] identify several possible sources of intrusiveness in aviation, including poor cable management and design, recalibration issues of signals, and discomfort caused by head mounted eye trackers compared to stationary eye trackers. Ref. [46] suggest that current EEG set ups based on the 10–20 electrode placement are unsuitable for daily fatigue measurement use due to the annoyance from electrode placement, and sensors such as wearable ECG offer better solutions due to requiring only a few electrodes on the chest. Intrusiveness is a potential risk when leveraging sensors for operational contexts, thus they need to be closely matched to operator requirements and working conditions.

#### 3.2.2. Size and Weight

When developing and evaluating wearables, researchers and developers should attempt to make the devices as small as possible. Wearable devices need to be compact in size to fit well onto the human body [19]. Furthermore, the size and design of the device in terms of ergonomics should mean human movement is unrestricted [22]. Whilst valid and reliable, some roofers in [47] noted that a wearable heart rate monitor on the chest was uncomfortable and restricted appropriate flexion and extension of the trunk. Whilst there is no formal guidance on what is too heavy for a wearable device, ref. [48] found participants did not notice a wearable device attached to a belt of approximately 500 g for a total of 30 twenty metre laps, even though the device could be felt in a localized position. Very recent devices, such as the Smarting EEG device, only weigh around 60 g. Furthermore [49] developed a pair of glasses for fatigue detection that only weighed 30 g. Thus, if a localized weight of 500 g is unnoticed over ambulant conditions, it is unlikely that wearables under 100 g will cause an intrusion or distraction. Wearable devices should not impact human mobility, as this may have negative consequences in operators who work in environments such as search and rescue, mining, and firefighting.

#### 3.2.3. Ease of Use

For wearable devices to be useful for both researchers and users, they need to be easy and simple to use. Both [8] and [10] note the importance of ease of use, with the latter suggesting a device should be simple enough for everyone to use. Ref. [8] note that devices should have a high degree of implementation, i.e., they should be easily integrated into a current system, e.g., fatigue management plan, and organizations wishing to use wearable devices should also consider any training needed to use the devices. Ease of use does not apply simply to the devices themselves, but also the outputs they produce. Devices should lead to simple and understandable outputs for everyone. For example, many fatigue management devices, such as the Optalert Eagle (wearable infrared sensors embedded in a pair of glasses), and the Life-band (wearable EEG device) by SmartCap, produce simple scales which allow an individual to understand their current level of fatigue. Augmented Reality (AR) devices which require interaction, for example the manipulation of a 3d OR 4d model, should use natural user interfaces [50], i.e., systems which use human behaviour or gestures to manipulate information and systems. A common example of this is the zoom function found on smart phones. This function works using two fingers, with increasing proximity of two fingers zooming the lens, while decreasing the proximity between the fingers zooms out. This is a very simple gesture and can be achieved with a single hand, allowing ease of use.

Additionally, devices should be easily integrated into the current system, with set up and calibration times kept to a minimum. As an example, the Optalert Eagle time requires no set up or calibration, and an operator can wear the Eagle device and get instant feedback on their level of fatigue. Other devices, such as the Tobii pro glasses 2, require calibration, with more detailed information in the user guide section. Ideally, devices should be “plug and play” in the app and device and be installed and ready to use without further training or calibration. If calibration is required, it should be quick, generally reliable, and simple. A device which requires greater than 15 min of calibration, multiple attempts and several steps to achieve calibration is unlikely to be adopted by operators. For example, for EEG calibration, ref. [51] recommend a calibration time of 30 s to occur before and after recordings. This is a relatively short duration, which is not unreasonable at the start of every day.

Developers and researchers should create and evaluate devices which are easy to use in terms of interactions, outputs, and training and set up times. Devices should be simple enough to use so that individuals without any great level of technical knowledge can utilize them effectively.

#### 3.2.4. Acceptance

A key variable as to whether an organization wishes to adopt wearable devices is that of acceptance. Acceptance refers to the degree operators will accept the device as a measure of the chosen metric, e.g., workload, stress, or fatigue [8]. Acceptance may also simply be viewed as whether the operators will use the technology [10]. A large barrier to the acceptance of wearable technology relates to concerns about privacy. For example, ref. [52] found that truckers are highly resistant to any sort of camera in the cabin, as trucking is typically a private occupation, and truckers enjoy the privacy of their cabins. Similar concerns were found amongst construction workers, who were resistant to having their location monitored due to fears of idle times being discovered [53]. Furthermore ref. [54] found that health professionals across a multitude of occupational settings, including energy, oil and gas, academia, research, manufacturing, and food processing, cited their greatest concern about employees’ adoption of wearable devices for health monitoring was privacy. They did, however, state that they saw the potential for wearable devices to improve health and safety in the workplace. When discussing the use of devices with operators, it is crucial that confidentiality is preserved and outlined to be of the utmost importance. Operators may be resistant due to fears of performance monitoring and the use of wearable devices to assess salary expectations. Wearable devices, especially those which measure physiological signals in the body, should never be used for performance management in the sense of deciding what salary an operator should receive. Firstly, whilst many devices provide decent evidence supporting their validity and reliability, the information gained says nothing as to whether the individual is “good” or “bad”. Being stressed or fatigued at work does not mean someone is bad or good at their job, and all that is being confirmed is that a psychophysiological state is present within the operator, which can affect their behaviour, resulting in negative outcomes such as an increased risk of accidents. Wearable devices used in operational contexts should be used to improve operator health and safety through either monitoring vitals or increasing awareness and communication. They are unsuitable for any other context.

Furthermore, some operators may question the use of physiological monitoring to assess psychological states as a valid method of fatigue monitoring. In such situations, it is important to stress the benefits regarding health and safety such devices can have. It is unlikely all operators are interested in the validity and reliability of devices to the same extent as researchers. Hence, it is important to focus on how devices will improve their health and safety and, by such means, overall job satisfaction and performance [9].

#### 3.2.5. Wireless Communication/Connectivity

Wireless connectivity is important for wearable devices in order to transmit data or outputs accordingly. Ref. [19] state that wireless connectivity allows continuous monitoring of human behaviour as well as removing any wires that may obstruct movement or make wearing the device uncomfortable during ambulation Connectivity through Bluetooth or Wi-Fi offers easy transmission of data to output devices. The advantage of Bluetooth and Wi-Fi as choice of connectivity male them a widely used solution in many cases, especially in the western hemisphere. For example, 84% of adults aged 16+ in the UK owned a smartphone, as of 2022 (Office of National Statistics, https://www.ons.gov.uk/aboutus/transparencyandgovernance/freedomofinformationfoi/percentageofhomesandindividualswithtechnologicalequipmen, accessed on 5 July 2024). Wi-Fi and Bluetooth have become standard on any smartphone, thus choosing these wireless protocols to transmit data means that outputs are accessible to a large amount of people. Ref. [55] found Bluetooth connectivity to drew less power than Wi-Fi connectivity, albeit the difference only being approximately 3%. However, it is likely that Bluetooth connectivity will draw even less power with the introduction of low energy Bluetooth or Bluetooth 4.0 in 2010, with this requiring only half the power compared to previous devices prior to 2010.

One of the issues with wearable devices stated by [19] and [21] is the issue of packet loss and poor data transmission due to ambulatory movement when utilizing Wi-Fi and Bluetooth connectivity. Movements, such as arm movements, as well as position, can affect latency of data signals, causing them to become unreliable. For example, ref. [56] found that a wearable EEG device stopped transmitting data when standing up or sitting down, due to traffic overload, as well as signals fading during movement. The authors suggest using a context aware medium access (CA–MAC) which can handle reflection, refraction, and absorption of data by the human body during ambulatory contexts. CA–MAC improves latency through dynamic scheduled-based and polling-based slot allocation. For a more in-depth discussion of the architecture and challenges of wireless connectivity, please see [56].

When developers and researchers are evaluating and developing wearable devices, they should consider having data outputs through Bluetooth or Wi-Fi, given how common these tools are. Furthermore, they should consider the reliability of data transmission between the device and the output source, whether it be a smartphone, tablet, or monitor. A device which stops producing reliable transmission during ambulatory conditions is very limited in its applications, hence solutions such as CA–MAC should be considered in the design phase.

#### 3.2.6. Individual Differences

In their systematic review of usability issues for wearable sensors, ref. [20] highlight the difficulty of creating universally good user experiences for all users. Often, designers and developers must make considerations about the wearable interface itself, e.g., whether it has a screen, the size and color of the font used, the button locations and size. Such choices may result in usability issues due to the user’s background, such as age and the presence of any disabilities [20]. Whilst universal guidelines and user-centered design can be used to offset, for example, usability issues resulting from poor readability, ref. [57] state that it is often impossible to create a device that is accessible to every population [20] suggest, at all stages of the product’s life cycle, from inception to commercial release, that usability testing should take place across specific demographics, age, gender and disability, to track usability issues across the device’s development lifecycle. Additionally, designers and developers could consider a modular approach in the wearable device to customize and try and broaden usability for specific subgroups. For example, in the addition to an LED based interface on color, duration and number of flashes to indicate when the device is charging or recording, it is possible to embed either auditory notifications within the device or a partner app to assist visually impaired users.

#### 3.2.7. Wearability

Ref. [13] coined the term wearability as the relationship between the device and the human form. The authors highlight the importance of wearability when designing digital technologies and smart garments and recommend several design principles for developing good wearability in products, including body placement, body fit, ease of movement, size, and comfort aspects, such as materials flex and stretch and thermal comfort, as well as aesthetics and long-term use.

Whilst [13] offer excellent guidance that sets the case for wearability, it is possible that psychological factors also impact wearability. Ref. [13] suggest that devices should be worn on the chest as opposed to the wrist, participants can prefer wearing devices on the wrist to the chest, due to improved comfort, social acceptability, and wearability [58]. The comfort scores and wearability between sensors were likely conflated to a certain extent due to the types of chest mounted device. Two out of three of the chest mounted ECG devices used disposable electrodes, whereas all the wrist-based devices used reusable electrodes. Disposable electrodes are known to be painful to remove and to cause skin irritation over a period of time [59] thus it is to be expected that, due to the device selection, the chest mounted ECG would score poorly in terms of comfort.

Interestingly, the third ECG device, the Polar H10 chest band, commonly used in research and developed specifically to be worn during exercise, also scored poorly in comfort ratings due to the band being tight round the chest, despite the Polar H10 using flexible electrodes. This suggests that familiarity and user end-goals affect perceptions of wearability and comfort. Ref. [60] suggests that social acceptability plays a large part in how wearable people score devices. For example, wearing devices that are unflattering, require unnatural, large and explicit gestures, as well as placing the sensor on non-neutral locations (areas of the body associated with sex, pleasure and elimination of bodily waste) all reduce the acceptability of the sensor. Ref. [60] identifies both the upper chest and wrists to be good placements for wearable sensors, yet ref. [58] found people even the wearing of chest mounted devices under clothing to be less socially acceptable than wrist worn devices [61,62]. Furthermore, ref. [63] define wearability as “the degree to which sensory stimuli generated by a worn object intrude into the wearer’s conscious attention, and we suggest that this intrusion has cognitive consequences for the wearer” p. 302. [63] go beyond [13] and directly link wearability to cognition and intrusiveness (discussed earlier), in that wearable sensors should not intrude on a user to create negative cognition, such as distraction, which is of even more importance when considering the application of sensors in a safety critical environment. Ref. [63] highlight that wearability is comprised of comfort, body schema and peri-personal space.

According to [63] comfort is comprised of five components: pressure, texture, thermal balance, moisture transport, and freedom of movement. A device that constricts due to excess pressure is unlikely to be adopted by users. Texture is a phenomenon that is felt dynamically. Unless the textile or material is very rough, the texture may resemble a stroke, scratch or flutter, which may become painful over time due to rubbing, and can be exacerbated through poor fit. Thermal balance relates to the degree to which a device’s pressure and texture is experienced during hot and cold temperatures, as well as body heat. This is closely linked with moisture transportation, the ability for the device or sensor to modulate the feeling of wetness caused by sweat or external moisture. When a device is wet, the texture and pressure will often change, thus increasing discomfort.

Freedom of movement is the degree to which a device or sensor prevents or impedes natural body movement and locomotion. In general, most wearable sensors are designed with this in mind, though excessive plastic or rigid materials may impede the wearer [63]. Hence researchers and designers should place some import on an assessment or judgement of wearability when building and selecting devices as, without wearability, it is likely that users will not wear devices for a sufficient duration, or at all, to obtain any meaningful benefit, no matter how precise the sensor is.

#### 3.2.8. Power Consumption

The most common power source for wearable devices is a battery. These will either be:(i)a non-removal battery, as seen in devices such as the Life band EEG device or Hexoskin, which is charged by the universal serial bus or micro-universal series bus, or(ii)AA or AAA batteries [64,65]).

Depending on the device, battery life can vary considerably, e.g., the Smarting EEG can run up to five hours prior to charging, whereas the Hexoskin heart rate monitor can run up to 30 h. Designers and researchers should look to maximize battery life as much as possible. For example, whilst the Smarting device does offer considerable time for recording data, five hours may be insufficient to capture the intended event, such as with fatigue during a 12-h shift. For example, designers and researchers should consider factors like ultralow power blue tooth as a communication device, as communication systems consume the most power [19]. Building micro-processing units using Advanced Reduced Instruction Set Computing Machine Architecture (ARM) is a power efficient solution, as ARM units use fewer transistors, registers, and circuits compared to more complex computer architecture used by companies like Intel and ATI [66,67]. Furthermore, micro-processing units using ARM architecture are developed with short and regulated length code, which further reduces power consumption. The computing carried out by these units is simple, compared to the code present within an intel processor, which is extremely complex. Thus, [66,67] both recommend ARM units in wearable devices to reduce power consumption and promote battery life.

Developers should always strive for the longest battery life possible, as not only does this improve the usability of the device but makes it a considerably more attractive solution to buyers [19]. Using low energy engineering where possible in the device should help achieve the longest use times possible. Another important consideration for developers wishing to sell commercial devices is the type of battery to use. Often, wearable devices favor a lithium-ion battery due to their small form and rechargeable nature, compared to traditional alkaline batteries. In the future, developers and researchers may decide to use motion-driven energy harvesters in place of batteries. These harvesters are micro-generators which harvest energy from ambient temperature, human locomotion, and vibration [68,69] Additionally, motion driven energy harvesters may also use piezoelectricity generation, in which the force of mechanical stress, such as a ball bearing hitting plates due to movement, stress or vibration, causes voltage to be created and harvested from the impact [70]. Thus, motion-driven energy harvesters may prove a smart choice, as they increase the longevity of the device and measurement phase, but also requires no charging or swapping batteries, which would prove an attractive feature for many buyers. This would also mean no loss of data due to practically unlimited continuous measurement.

### 3.3. Bridging the Gap—How Design Can Influence Measurement and Usability

Understanding how measurement and usability may impact the sensor’s performance and user experience is important, but researchers and designers should also be aware of how design can help improve measurement and usability, not just for the operators themselves, but also for optimization of devices to be used outside of laboratory contexts, such as in the field. When purchasing or developing a wearable sensor for physiological measurement, there are several design principles to consider relating to interaction and user experience, as well as hardware considerations, to improve usability and measurement [20].

#### 3.3.1. Toughness

Wearable devices should be tough enough to withstand the rigor of everyday life. When one uses a wearable device to assess the wellbeing of operators, often these devices will be used over the course of shifts which can last anywhere between 6 and 12 h. The Office of Rail Regulation notes that it iss critical to manage operators in safety critical environments effectively, and reliable monitoring strategies are part of that. If a wearable device cannot last more than a few months without degrading, it is likely that they will not be perceived as useful due to the need to constantly buy new equipment. Ref. [10] state that wearable devices should be able to cope with heavy usage. Whilst it may be unrealistic to demand a device that lasts a decade, a wearable sensor which lasts several years before degrading would certainly be reasonable. Ref. [22] suggest that developers and researchers should consider the context the device will be utilized in to maximize their durability regarding abrasion, impact, flexion, and humidity. In addition to the points stated by Motti and Caine, devices used in environments where dust and sand are highly present should be reinforced so that devices are not contaminated. Dust is a particular worry for electronic devices, as dust present within electronic devices promotes absorption of water molecules from the atmosphere, which leads to failure of electronic devices [71]. In a similar manner, high levels of salt present in the air or water may also corrode electrodes and metal components if the device is not appropriately sealed. This suggests that sensors should be designed not just from a user centered approach, but also with environmental contexts in mind. For example, the same sensor used in a construction environment is likely to have more demands on toughness than in a control room, regarding how the device is sealed in relation to dust or sand.

Furthermore, ref. [72] suggest that textile components, including textile electrodes, should be resistant to abrasion and washing. During use, textile components stretch, shear, flex and rotate, and then suffer from soaking, rinsing, drying and spinning, as well exposure to cleaning detergents. All of these processes can weaken textiles and have an adverse effect on electrode impedance [72]. Thus, researchers, but more so practitioners and designers, should consider both material type and electrode knit when trying to optimize toughness in textile components. Ref. [73] comparing several differing textiles, stitched silver coated yarn electrodes of various stitch patterns and seam lengths using a W6 model N1800 sewing machine. Ref. [73] found that a stitch pattern of backstitch plus elastic blind stitch with a seam distance of 3.0 (Machine specific units) showed excellent preservation of impedance after 30 wash cycles and 1000 cycles on the Martindale abrasion test, compared to all other stitch variants. Furthermore, the electrode, which was specifically designed to be low-cost and easy to create, performed closely to that of an industrial electrode used as a baseline. Ref. [25] echo Zaman’s sentiments, in that textile components should be machine washable, resistant to stretching and abrasion and have sufficient durability to be worn on the body without becoming damaged.

#### 3.3.2. Water Tolerance

Devices need to have some degree of water tolerance, to either sweat or environmental moisture from humidity or rain. Ref. [19] state that wearable devices should withstand variation in environmental conditions in relation to temperature, moisture and water droplets. This is important, as devices that require the control of variables like temperature and moister to preserve accurate measurement are not suitable for all operational settings, and may be restricted to enclosed settings, such as air traffic control. Ref. [74] note that humidity and ambient temperature may affect accuracy of sensors, with some thought needed towards the types of environment sensors will be used in. Ref. [25] suggest that any textile electronics should be coated or have an interface layer that acts as a barrier to prevent any water penetrating the electronics if they are not already waterproofed. If a wearable device is only accurate or comfortable in moderate climates, such as the UK, as opposed to warmer humid climates, such as central and northern south America and southeast Asia, as well as being unsuitable for water-based work environments, then the measurement potential and usability of the sensor becomes severely limited.

#### 3.3.3. Interaction Methods and Feedback

Wearable sensors can vary both in how users interact with them (buttons vs. smartphones) as well as the types of feedback given, e.g., auditory, force, visual, etc. Ref. [20] summarise that interaction methods should be closely matched to the user and, if needed, even customised to users’ needs, e.g., setting the threshold for touch detection, ensuring gestures are not unnatural, such as swiping around the head/face, and ensuring voice interfaces work well for the intended context, such as not being influenced by accent or external noise.

Users may also prefer using a smartphone interface due to increased iconography and text size, familiarity with using their own smartphone’s touch interface, with improved visualisations and clarity, over an interface located solely on the wearable sensor [20]. This offers both benefits and challenges. For users who may have impaired vision, a smartphone interface may be preferred due to the ease with which smartphone interfaces can be customised regarding font size, font type, brightness, etc., and, if a user is already familiar with a smartphone, then using the smartphone to interact with the device may create a more positive and familiar experience than purely using the device itself. Furthermore, smartphones can convey more information to the user, for example showing the word ‘recording’ versus a certain colour or flashing LED may offer more clarity and assurance to the user that the device is recording. Here, the user does not need to learn the rules of the device’s interface, rather relying on what they already know, i.e., that the word ‘recording’ indicates data is being recorded, reducing the cognitive load and ambiguity for the user. However, requiring a smartphone interface may limit users who do not have smartphones, may not be allowed smartphones for any reason and, if it requires an actual smartphone, do not always have their smartphone present for interaction. This then leads to the additional need to ensure that the smartphone lasts for the recording session as, if the only way to interact with the device is through a smartphone, one would then become conscious of the smartphone battery being drained if it must continuously interface with the device.

The interaction method is an important consideration for how easily one’s goals when using the device can be matched. For example, both the Empatica E4 and Equivital EQ-02 require buttons on the devices themselves to be pressed to mark timestamps. This feature works well in contexts such as exercise, but for the monitoring of safety critical environments, and especially experimental work, it would be better if an experimenter could timestamp the data without having to place the burden on the participant. One device that has removed this burden, but maintains device interaction with the wearer, is the Astroskin device. Rather than press a small rubber button, the wearer simple strikes the Astroskin processing twice, a movement akin to patting one’s own chest; the device then vibrates to indicate to the wearer that the timestamp has been marked. Here, the Astroskin leverages haptics to make the interface simple and easy to use. There is no need to look at the device and focus attention away from the task. The feedback is a vibration rather than some visual indicator, which also preserves attention, and works in the same way regardless of light levels. Certain LED colours, such as orange and yellow, may be hard to see in certain bright lights, and using a smartphone intrudes on the task at hand for timestamping purposes, which results in a final benefit of improving safety through reducing the physical and visual burden on users in trying to timestamp data.

#### 3.3.4. Onboard Processing vs. Secondary Device Processing

Ref. [20] state that wearable sensors should try and limit their reliance on using smartphones and apps to stream and collect data. They state that secondary devices create increased stress and workload when interacting with wearables, resulting in a more negative user experience to that of devices that do not require additional devices to save, stream or see the data. For most consumers, this is the ideal and, in normal everyday settings, having a singular device is likely to be the preferred choice compared to having a secondary device. However, there is an argument to suggest that in some contexts, such as online and real time data processing and analysis, allowing the data to be exported or streamed to a secondary device in real time is advantageous. Running models to identify psychophysiological states is often computationally intensive, but these models can run on current smartphones. For example, ref. [75] developed a fatigue detection system for drivers which combined a wearable EEG and a smartphone running classification models. Refs. [76,77] state that leverage machine learning models in real time is a computationally demanding process. Whilst the gathering and storage of initial datasets could be saved on the primary device, it is likely that sensors cannot be as discrete and meet requirements regarding comfort whilst possessing enough processing units to run the model in enough time to meet the users’ goals, either from a CPU, GPU, Field Processing Gate Arrays or Application Specific Processing Units [77]. By shifting the burden of cleaning, analyzing and outputting data from desired models to mobile devices, the sensing unit itself can be optimized regarding hardware, battery life, comfort and fit, in that by removing some functionality from the sensor itself, its potential as a sensor can be maximized.

Furthermore, many researchers like to visualize the data before it is collected to assess whether the signals they capture are as noiseless as possible. Many sensors do not have screens, thus using a secondary device to stream and visualize the data can be useful to check data quality before collection occurs. The Astroskin is a good example of this. The Astroskin device carries out all the data recording and processing, whilst one can use a smartphone app to view the data, akin to a window.

Using an additional secondary device to store and capture data can present risks. There may be software issues as well as issues with connectivity and latency [20]. Furthermore, if the secondary device runs out of battery or crashes, then the model itself will not run, which may result in safety issues if the user is relying on the system beyond experimental contexts. This could be avoided depending on the type of secondary processing device. If the device is a laptop or computer embedded into the environment and plugged into a power source, it is likely that battery issues will not be a problem. If the secondary device is a smartphone, then users may be cautious about just how long the two devices can be used together without the battery running out, as the smartphone would require to constantly receive, analyze and output data, which is likely to be a large drain on the battery. An alternative is to compress the models to run on very computationally weak devices, like Arduinos and the Raspberry Pi [78].

If the wearable sensor has the bandwidth in the design to accommodate a processing unit onboard at a reasonable cost [79] without compromising how the sensor fits on the body, data quality and long term comfort [20,43], then this is what designers or developers should strive for; however, if additional processing is needed or would provide a more cost-effective solution, then designers and developers should consider leveraging a secondary device to handle the burden of the processing needs.

#### 3.3.5. Modularity

Modularity, i.e., to be able to change, remove or replace parts of the device or sensor, should be a consideration for both designers and researchers. Ref. [23] suggest sensors that can be placed on multiple body locations, e.g., in their case, EMG devices should be modular in that different electrode types can be optimised to body locations. A similar application could also be applied to, e.g., a PPG device that can be worn on the wrist, or, if that is not possible, worn on the forehead.

A further example comes from Plux Biosignals (https://www.pluxbiosignals.com/, accessed on 23 May 2024), which offers several different sensor kits, ranging from prototyping and experimentation to higher grade research kits. The Bitalino range is relatively cheap, and many different sensors can be connected to a singular processing unit, such as ECG, PPG, EDA, EMG and, EEG. This offers flexibility in what signals researchers can collect, and they may want to collect data using some or all of the sensors, depending on the research question. This creates a cost-effective solution in which researchers only need one central processing unit to actually sample and collect the data, and purchase an array of sensors, rather than purchasing several different devices. This also means that the device is simpler to repair, rather than having to send back an entire device, and if one can diagnose the offending piece of the modular sensor, the broken or problematic piece can simply be replaced. In a similar vein, any future new sensors can be simply integrated into the processing board the researcher already owns, rather than having to purchase an entirely new device if they wish to collect new signals of interest [80].

Furthermore, designers of devices should consider allowing for some flexibility in device placement and where it interfaces on the body. For example, one could have a singular processing unit, which slips into a smart shirt that is worn to measure ECG, and the processing unit can then be placed on the leg through a strap, which measures EMG of the upper leg muscles, and finally can be integrated into an EEG cap to be the processing unit for an EEG system.

By affording some modularity across devices, different types of research paradigms can be assessed, as well as the usability of the device improved. Consider a final example: individuals may differ on where heart rate data is collected from, the two most common locations being the wrist as a PPG device or a chest mounted ECG. Imagine a singular device that can be ongoingly augmented to collect data that is of equal or very similar quality, but placed on an area of the body dependent on user preference or need.

Ref. [20] also note that, by allowing modularity in device aesthetics, users can change the look and feel to meet their needs. Devices like Fitbit currently offer this, where users can change case colors and wrist strap designs. Whilst this would be desirable, it is possible that operators in safety critical environments may not place as much stock on how a device looks compared to consumers, who want devices to fit seamlessly with their own outfits and aesthetic. Whilst there is very little research exploring modularity in sensors, ref. [80] suggest that having modularity in wearable sensors for medical assessments improves versatility and assists with maintenance and assembly. This sentiment also applies towards more extreme environments, like offshore wind. Whilst one would not expect operators to carry around spare sensor parts, having a stock of additional electrodes or a PPG/ECG/EDA device could be easily interchanged when necessary or when needed depending on environmental demands.

Modularity is not an easy design consideration to integrate, as most sensors require complex holistic systems to achieve reliability, a good level of signal cleanliness and usability. However, both designers and researchers should, where possible, try and integrate some level of modularity, when this offers value both to themselves and their users.

#### 3.3.6. Type of Electrode

Beyond light sensors, like PPG, most current commercial devices rely on electrodes as the measurement medium. Ref. [24] distinguish between the three main electrode types, traditional wet gel electrodes, dry electrodes and textile electrodes. However, each has their own advantages and drawbacks, thus researchers and designers may wish to spend some time investigating which is best for their purposes regarding measurement quality and overall usability for operators across various environments.

Traditionally, electrophysiological devices, like ECG, EDA and EEG, rely on wet gel electrodes to collect data. These electrodes work by placing conductive gel between the skin and the electrode; however, this process can be messy and is not practical for everyday use in consumer or occupational contexts. Furthermore, the electrodes are typically made up of silver/silver chloride (Ag/AgCl), which, whilst offering a stable signal and low skin to electrode impedance, can often be uncomfortable and after several hours can irritate skin (An et al. 2019 [59]). Thus, there is a movement towards researching textile and dry electrodes. These are more suitable for wearable sensors as they offer long term comfort, do not rely on conductive gel, tend not to irritate the skin and do not require preparation. The downside of these electrodes is that they often lack the same signal stability as Ag/AgCl electrodes, though recently several studies have identified manufacturing advancements in electrode design that have brought them close to Ag/AgCl.

##### Textile Electrodes

Textile electrodes tend to be either a combination of textiles and metallurgic materials with metals woven into the fabrics themselves, or fabric and textiles coated with a metallurgic compound [59]. The other type of dry electrode that exists are electrodes that require no conductive gel in the case of EDA and ECG, typically a small spiked or pin type electrode that has direct contact with the skin with regards to EEG [81]. Currently, there is also a rise in what are known as non-contact electrodes. Non-contact electrodes are electrodes directly attached to the circuit board, then covered with a solder mask to reduce noise and artifacts in the signal [82].

In their recent review of textile electrodes, ref. [83] suggest that textile electrodes need to be flexible, comfortable and not irritate the skin, have a good signal quality, be washable and be durable, so that repeated use and washes do not degrade the material and signal over time. They also suggest that the textiles should be easy to use, and not require any specialist training to set up. They outline that these are the necessary parameters needed for a textile electrode to be successful. The author would also expand this to the application to safety critical operations, in that humidity, dust, vibration, excess movement and operations do not degrade any aspects of the textile electrodes. This also applies to temperature, sweat and breathability. A textile electrode that creates issues around odour, thermal discomfort, itchiness, etc., is likely to intrude on the user, which is especially dangerous for the operator. Therefore, any device needs to be assessed against these criteria, if textile electrodes are the method used in the sensor.

##### Dry Electrodes

In a similar vein to textile electrodes, dry electrodes offer the potential to collect data over several hours without the need to reapply conductive gel as seen with wet electrodes. There has been a relatively large movement in the EEG community to develop dry electrodes to improve set up time, comfort and longitudinal [82]. Additionally, dry electrodes have the ability to bring electrophysiological measurements to the consumer market [84]. Thus, it is of interest for researchers and designers to consider the challenges associated with leveraging dry electrodes, despite the benefits.

Ref. [84] compared three EEG electrode types: passive wet, active wet, and active dry. They assessed each type’s ability to produce a usable EEG signal during an auditory oddball task and to detect the P3 ERP response. Results from 300 randomly selected trials showed that active dry electrodes had significantly more noise than passive wet and active wet electrodes. All electrodes detected the P3 ERP response at about 380 ms after stimulus onset, but active dry electrodes had the noisiest signal. Further analysis revealed that active dry electrodes required significantly more trials (500 standard to 125 target) to reach statistical significance compared to both wet electrode types, which needed fewer trials (125 standard to 35 target). Mathewson et al. concluded that, while active dry electrodes can detect the P3 ERP response, they generate more noise and need more trials for the same statistical power as wet electrodes.

To further improve dry electrodes, ref. [85] suggest the use of nanomaterials to construct electrodes, because they have greater flexibility than current dry electrodes, resulting in increased skin-to-electrode contact due to the ability to fit the uneven and curvilinear surface of the body, and with reduced motion artifacts because of this. Ref. [85] highlight in their review how carbon nanotubes, also known as CNTs, can perform equally as well as traditional wet gelled electrodes for ECG measurements. Additionally, they describe a study where a CNT ECG patch was worn for 7 straight days and showed no signs of dermal irritation or toxicity to the skin. Similar findings were observed with graphene clad electrodes in that they produce an almost identical ECG waveform to that of wet gelled electrodes [85].

They conclude that dry electrodes based on nanomaterials offer a promising solution to improving longitudinal signal stability and removing motion artifacts, if manufacturing and developing these electrodes is relatively simple and cost effective. CNTs, however, may only be effective for ECG electrodes, as opposed to EEG electrodes. In their review of dry electrodes for EEG measurement, ref. [86] point out that, whilst CNTs have potential in the manufacturing of electrodes, currently they can only be used in the form of microneedle arrays. Microneedles are tiny pins that do not penetrate the epidermis and are not felt by the individual, and thus are technically classed as non-invasive. Due to the lack of skin contact from microneedle arrays, they require high pressure to create and maintain good impedance, thus are not particularly comfortable for the wearer.

CNT has the potential to be toxic to humans and can only be manufactured to a very short length due to their mechanical properties, thus are not suitable for individuals with long/thick hair as they cannot penetrate the scalp appropriately [86]. Whilst nanomaterials appear to offer a good solution for long term comfortable electrodes, not all designs are equal and thus they should be tested in various formats, e.g., patch vs. microneedles, to ensure they are being utilized in the correct way. Both [85,86] note that nanomaterial electrodes are an emerging topic in physiological research, so it is likely that in a few years they will become even more prevalent and, if manufacturing processes can be cost effective [87], they will drastically improve both research and consumer grade electrodes. Ref. [88] demonstrate the possible benefits of nanomaterial electrodes leveraged in the correct way. Silver ink was printed onto polyethylene terephthalate (PET) substrate to create a flexible dry electrode. The dry electrode produced a better ECG signal than that of over gelled wet electrodes and showed fewer motion artifacts during ambulation.

Researchers and designers have many parameters consider, such as the material, location pressure and size of dry electrodes and, whilst they are starting to be used in some commercial devices, such as the Empatica E4, with success, there are still questions to be asked about their signal-to-noise ratio due to, at times, poor skin to electrode connectivity. However, recently nanomaterials appear to offer a solution in that they can fit the body to a much better degree, enabling measurement as good as or even possibly better than traditional wet gelled electrodes. Novel methods, such as the magneto-cardiogram [89] and ballisto-cardiography [90] will hopefully offer alternatives to electrodes and associated issues as they mature.

#### 3.3.7. Cleaning and Maintenance

Wearable sensors over time will experience bacteria and dirt building up. This can impact both the measurement of the device, e.g., degrading/scuffed electrodes, as well as usability, e.g., comfort. Thus, they need to be easily cleaned and maintained [25]. The ability for textiles and textile electrodes to be washed again and again and not degrade after multiple wash cycles is defined as washability [72]. Ref. [73] suggests that clothing-based sensors should be machine washable without degradation from electrical impedance and the fit of the garment, even stating that this should be a mandatory consideration due to the fact that sweat corrodes electrodes when they are worn frequently. In addition to silver coated yarn, ref. [91] found that carbon nanotube threads, which behave in a manner allowing them to be sewn, did not have signal degradation after several wash cycles. Refs. [73,91] demonstrate that it is possible to create washable textile electrodes that do not degrade with wash cycles. Washability should be strived for, as researchers and designers no longer have to be responsible for manually cleaning the electrodes, rather they can just be placed into the washing machine, which has the added benefit of saving time. More generally speaking, all wearable sensors should be easy to clean, e.g., it should not be difficult to clean any electrolyte gel from electrodes. For example, the Easycap EEG (https://www.easycap.de/, access date 21 October 2019) recording cap is made up of textile and metal components, all of which, except the connector, which is used to connect the cap to the processing device, can be submerged in water and scrubbed easily with a toothbrush. Many wrist-worn devices, which use straps, can be wiped down and cleaned easily. Some even boast designs where the screen/processing unit can be removed from the strap entirely, which allows a new strap to be placed, and the device to be cleaned to an extremely granular level.

This goes hand in hand with maintenance. Devices should be simple to clean and easy to maintain. As previously mentioned, modular devices assist in maintenance in that they can be broken down into their components to clean, as well as improving maintenance of the parts. Most devices, however, are not designed to be maintained by the user, but rather must be sent back to the company to be fixed. There are some exceptions to this, however. The Empatica E4 does not have integrated electrodes, but they are popped onto the device, meaning that, if the electrodes become corroded, then can be easily replaced. Additionally, companies, like BITaliano and Open BCI, offer products where the user does the assembly themselves, which would allow someone of little knowledge with some guidance to possibly repair the units/components, since the entire processing board is not contained within a housing unit. For example, take the Open BCI Ganglion board; some of the pins on the back of the ganglion boards, which connect to the cap or electrodes, could become bent out of place if a user is not careful in disconnecting the recording cap or electrodes; before the user has to send the board back, or purchase a new board, they could bend the pins back and see if that fixes the issue.

#### 3.3.8. Sampling Rate

Sampling rate refers to how many times per second a frequency is sampled, e.g., a 60 hz sampling rate would mean that each second of the recorded signal would contain 60 samples per second. When selecting a device to measure physiological phenomena, one should consider the time series of such phenomena, as well as the expected saliency of the phenomena, e.g., are we measuring two extremes, or several states that only differ gradually? Asking this question will help infer what type of sampling rate the device should have. Sampling rate is important, as a high sampling rate is more robust and is more effective in frequency-based analyses [92,93]. Ref. [26] suggest that the minimum sampling rate should be 100 hz, ideally between 250 and 500 hz. The author also suggests obtaining a sampling rate as high as possible in order to have the best chance to catch temporally sensitive events if that is a requirement of one’s research, e.g., regular changes in stress or cognitive workload may require a higher sampling rate than slow discrete changes.

There has been some research examining both the minimum sampling rate needed to catch physiological phenomena, as well as comparing wearable sensors to that of “gold standard” laboratory equipment, in order to assess whether lower sampling rates are sufficient in capturing enough reliable data to perform analyses. Ref. [94] compared 5-min samples of HRV data from a 64 hz PPG wrist-worn heart rate monitor to a 1000 hz ECG, with 56 participants. They assessed whether 10 HRV parameters derived from the data were significantly different between the ECG signal sampled at 1000 hz, the PPG signal sampled at 64 hz, and as several down-sampled PPG signals going down to a minimum 6.4 hz.

They found that, at 64 hz, the PPG signal and ECG signal were relatively similar, with only RMSSD and PNN50%, two commonly derived HRV parameters, being significantly different at 64 hz compared to all other HRV parameters. They suggest that, if researchers wish to use RMSSD and PNN50% parameters, they should select an HR sensor with a sampling rate greater than 64 hz, or the sensor is likely to be sufficient. Furthermore, they found that, when the PPG data was down0sampled, the minimum sampling rate with which one could obtain reliable data was 21 hz; however, the more likely number is 32 hz, as 21 hz differed on all but two HRV parameters compared to the ECG, where 32 hz only differed on 5 out of the 10 HRV parameters.

It is worth noting that even 32 hz differed on both RMSSD and PNN50% compared to the 1000 hz EEG, with only SDNN not being significantly different. Thus, from [94] findings, the author would suggest that even 32 hz is not likely enough to capture a sufficiently good quality signal when researchers are trying to measure psychophysiological phenomena, such as stress, fatigue and cognitive workload, as many HRV parameters, such as RMSDD, SDNN, and even PNN50%, have been identified as measures of the above-mentioned phenomena [95,96]).

Ref. [97] found similar findings to [94] when they assessed waveform features of the PPG signal at 240 hz, and down-sampled at various intervals to 10 hz. They found that, at 60 hz, the waveform was reliably similar to that of data sampled at 240 hz, suggesting that 60 hz is the minimum PPG sampling rate to be used in commercial applications. One caveat about [3] work is that it was specifically focused on waveforms for commercial applications in healthcare, thus their examination of minimum sampling rate to preserve PPG waveform is in the context of assessing disorders, such as arterial stiffness, cardiac output and vascular ageing. These metrics are likely to be nowhere near as time sensitive as psychophysiological constructs, like cognitive workload and stress, as well as temporal aspects of state fatigue. Therefore, their findings should be taken with slight caution, as PPG waveform may be altered due to activities related to the measurement of stress, fatigue or cognitive workload if additional signals or physical activity are required to measure the aforementioned phenomena [98].

### 3.4. Example Checklist

Whilst the framework offers the knowledge on how to select and assess a sensor, what would be a reasonable methodology? We propose a check list, which scores evidence against each parameter. This allows various checklists to be created, which can be used to easily compare different devices. See the appendices for the checklists and scoring instructions (Appendix A and Appendix A).

The checklist is normalized to give a score out of 100. Moreover, the checklist could be weighted desired, depending on operator need and data available. For example, the authors recently completed some survey work assessing the framework parameters in the energy industry. Respondents suggested validity, reliability, acceptance, cleaning and maintenance, and toughness as clear favorites. Therefore, one could weight these five parameters as more important than the rest if applying the sensor to operators in the energy industry. See the appendix for an example checklist. The authors suggest using the checklist ideally in a trial environment, which is backed up by peer reviewed literature, operational evidence and user manual information. In the absence of trials, relevant parties should complete the checklist in relation to industry standards/literature. The checklist is scored on a 0–5 Likert scale, which allows parameters that are not relevant or measured in the present context to be scored as 0. Below is an image of example data (Figure 2), randomly generated for the purpose of the paper, which shows two spider charts overlapping, comparing an ECG and a PPG for fatigue measurement. Higher scores indicate good performance on the parameter, e.g., a score of 5 indicates excellent validity. For many devices, one may prefer faceted bar plots or spider charts, as overlaying more than three devices may degrade readability.

## 4. Discussion

While we have attempted to classify the parameters to the best of our ability to the correct domain, we nevertheless acknowledge that arguments can be made to move some parameters from usability to design and vice versa. The usability and design domains have some degree of subjectivity as to whether they are truly principles of usability or principles of design. Measurement principles are easier to classify, and these parameters directly relate to how competently a sensor performs as a measurement tool, of course, in the context of the design choices, as has been discussed. Figure 3 shows the final framework and serves as the first attempt at a baseline for the assessment and selection of wearable sensors for psychophysiological monitoring. Our hope is that, in future, other fields and researchers can assist in building this framework, e.g., by evaluating different materials in relation to toughness.

Different contexts may lead to different parameters being prioritized over others. The overall goal of this paper is to create a framework that is sufficiently broad such that interested parties can gain an appreciation of measurement and usability and thereby understand how to review and assess these in their own research and development. Furthermore, this paper aims to help the reader understand how measurement and usability may be optimized through conscious design choices. Whilst it is hard to comment on the degree of importance of each parameter, in the writing of this paper, we would suggest usability is the crux as to whether people will adopt sensors in the long term. This is suggested by [20] and has been shown empirically amongst operators [99].

### 4.1. Study Limitations

The paper has several limitations. There is no empirical data to help support the inclusion of parameters. Whilst many parameters do stem from empirical studies, it would be useful to understand how operators perceive these parameters, and to which they impose most importance. This would differ based on many different variables, such as age, industry and perceptions of technology. In a similar vein, the paper does not include any case studies. These are often useful and can add clarity and context without being as theoretically complex as empirical studies. Finally, the paper only captures a static moment in time. The development of new and better wearable sensors is increasing exponentially, e.g., ref. [89] developed a Magnetocardiography (MCG) Wearable Sensor which does not rely on electrodes; thus the electrode type parameter would then become obsolete. The paper is written to be device agnostic, however, and there is merit in further examining parameters related to types of specific devices, e.g., ECG. Currently most psychophysiology devices rely on electrically-based measurements, with the exception of some eye trackers. However, recently there has been an increase in developing accurate and discreet PPG, fNIRS and even the development of the aforementioned MCG, which do not rely on electrodes at all. However, each of these devices has its own set of measurement complexities. Thus, there are likely to be device-specific parameters that we have not addressed but would be useful when considering the selection and development of such devices.

### 4.2. Future Directions

While this paper attempts to consolidate the literature from multiple relevant fields to create a holistic framework that can guide others, there is also room for further development. A similar review that is device-specific would be useful in highlighting nuances that are not device agnostic. Additionally, the inclusion of empirical research in a similar review would assist in supporting veracity, and that the parameters in question are valid and of concern to operators. Finally, a systematic literature review could help reveal possible parameters that may not have been discovered in our search, as well as understand which parameters may be most popular or occur frequently in the literature. Additionally, seeing practical applications of the checklist would help validate its use.

## 5. Conclusions

Wearable psychophysiological devices are more readily available than ever before due to advances in engineering, computing, and ergonomic design. With ever increasing numbers of devices available on the market, there is a need for a framework to guide the development and selection of these devices. This need becomes even more relevant when the aim is for the sensor to be used to measure some sub-optimal state amongst operators to improve safety. The authors hope that the proposed checklist offers a tool which researchers, safety managers and ergonomists can use and adapt to ensure that they are selecting the best possible device, underpinned by a sufficiently broad theoretical framework that aims to maximize deployment.

## Figures and Tables

**Figure 1 sensors-24-04589-f001:**
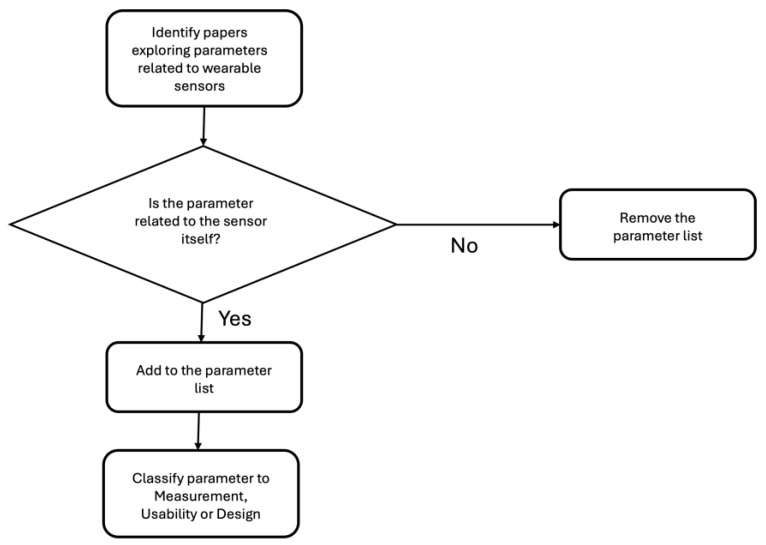
Flow chart of how papers were selected and integrated into the review.

**Figure 2 sensors-24-04589-f002:**
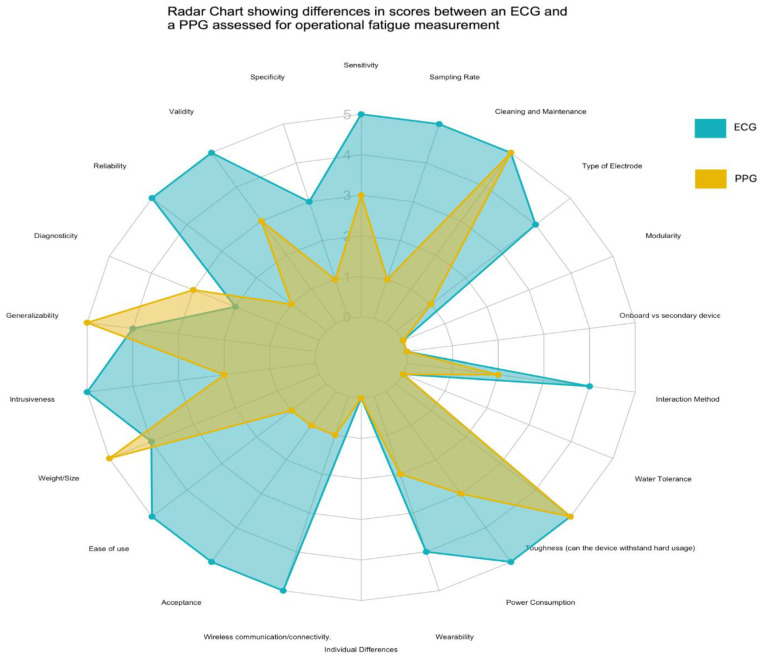
Example of a spider chart showing the difference in scores between an ECG and PPG devices across all parameters.

**Figure 3 sensors-24-04589-f003:**
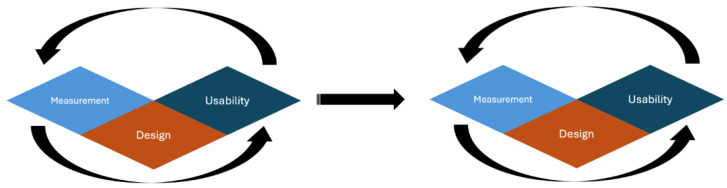
Comprehensive framework of principles which should be used to develop and guide evaluation of wearable devices which monitor human psychophysiology. Design is centered, as it represents the core tenet which maximizes both the “pillars of measurement, and usability”. The arrows indicate that sensors may be iterated through some or all of these parameters before a final version is deemed suitable for its context. Usability and measurement may interact with one another through iteration, with different priorities being focused upon, dependent on user needs and user feedback.

**Table 1 sensors-24-04589-t001:** Summary of all 22 selected parameters and the source of each parameter. All parameters were selected from peer reviewed articles and relate to either measurement, usability or design.

Parameter	Source
Measurement Principles	
Sensitivity	Dinges and Mallis [10]
Specificity	Dinges and Mallis [10]
Validity	Campbell and Fiske [17]
Reliability	Dinges and Mallis [10]
Diagnosticity	Boff et al. [18]
Generalizability	Dinges and Mallis [10]
Usability Principles	
Intrusiveness	Izzetoglu et al. [8]
Weight/Size	Kumari et al. [19]
Ease of use	Dinges and Mallis [10]
Acceptance	Dinges and Mallis [10]
Wireless communication/connectivity.	Kumari et al. [19]
Individual Differences	Khakurel et al. [20]
Wearability	Gemperle [13]
Power Consumption	Pantelopoulos and Bourbakis [21]
Design Parameters	
Toughness (can the device withstand hard usage)	Dinges and Mallis [10], Moti and Cain [22]
Water Tolerance	Kumari et al. [19]
Interaction Method	Khakurel et al. [20]
Onboard vs. secondary device processing	Khakurel et al. [20]
Modularity	Cerone et al. [23]
Type of Electrode	Ramasamy and Balan [24]
Cleaning and Maintenance	Islam et al. [25]
Sampling Rate	Camm et al. [26]

## Data Availability

Please contact R.H. for a database of papers the manuscript used as part of the Results section.

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
