# Peer review of "A Framework for Selecting and Assessing Wearable Sensors Deployed in Safety Critical Scenarios"

_sensors, 2024, doi:10.3390/s24144589_

Round 1
Reviewer 1 Report
Comments and Suggestions for Authors
The comments are as follows,
1. More tables are needed to illustrate or compare the main works of related refrence in a specific issure.
2. Enough recent references are needed to be surveyed.
3. The future works should inspire the students or researchers in wearable sensors,
4. The article does concern the fundamental issues in the measurement and usability of wearable sensors. Moreover, the writing of abstract, motivation, the surveyed issues, future works, conclusion are not professional yet. However, the wearable sensors does deserve to be surveyed based on the recent references.
Comments on the Quality of English Languagenone
Author Response
We would like to thank Reviewer 1 for the insightful and useful comments on how to improve the manuscript and hopefully bring it up to scratch for publication
- More tables are needed to illustrate or compare the main works of related refrence in a specific issure.
We do agree with this point, and if the manuscript was concerning primary data collection, we would certainly have included more tables. Whilst we actually did the opposite, and reduce the table and figure count, we did this to reduce the overall length of the paper. The team spent some time discussing whether more figured and tables would help improve the readability versus the length of the paper and decided against including more tables/figures, as the paper is quite long, and the text has been reduced and summarises the information in a concise manner.
- Enough recent references are needed to be surveyed.
We have updated some of the more outdated references, please see sections on Validity and Type of Electrode for some more recent references.
- The future works should inspire the students or researchers in wearable sensors
Included a checklist and a short sentence about students and researchers applying the checklist to validate it's usefulness.
- The article does concern the fundamental issues in the measurement and usability of wearable sensors. Moreover, the writing of abstract, motivation, the surveyed issues, future works, conclusion are not professional yet. However, the wearable sensors does deserve to be surveyed based on the recent references.
Team has edited the manuscript to tighten up and improve the overall writing.
Reviewer 2 Report
Comments and Suggestions for Authors
The paper is generally well-written and definately can be interesting for the readers of the Sensors journal. However, there are a umber of issues to be fixed before it can be accepted.
First of all, the paper is way too lengthy. It shuld be shortened at least twice. Some shortening considereations include:
Tables 2-4 are basically fragments of Table 1 but have parameter groups. Are they really needed? The groups can be included into Table 1.
Figure 2 doesn't seem to be important - it has several lines of descritpion that can replace the figure itself.
Some other issues:
Conclusion is not finished and has some working comment (did the authors read the paper before submitting it at all?).
The paper is about wearable sensors, so the wearability parameter seems to be excessive.
Type of electrode parameter only concerns specific sensors. What about optical wearable sensors? Maybe some others? The authors should either concentrate on wearable sensors with electrode-based information acquisition and indicate this in the title and abstract, or consider other types of connections.
The title speaks about "safety critical scenarios". Do the authors really concentrate on such scenarious? It doesn't look so.
Also, there is no methodology presented how to select sensors (as stated in the title) based on the presented assessment framework.
Author Response
We would like to than Reviewer 2 for the time taken to review the manuscript as well as the useful and thoughtful comments to help improve the quality and content, and hopefully achieve publication.
First of all, the paper is way too lengthy. It shuld be shortened at least twice. Some shortening considereations include:
The paper has been reduced. Not quite to the length you desired, as it was quite hard to keep enough information that was informative and provided guidance and thought. But we did manage to cut down several sections, most notably the sections concerning wearability and electrode type, however most sections did go under some for of editing to try and reduce the length.
Tables 2-4 are basically fragments of Table 1 but have parameter groups. Are they really needed? The groups can be included into Table 1.
Figure 2 doesn't seem to be important - it has several lines of descritpion that can replace the figure itself.
We have removed all the tables bar table one, but added in some labelling and formatting, as on reflection, several of the tables were redundant. The left Figure 2 in to highlight the iterative nature of the sensor selection and assessment, however are happy to remove it if need be.
Some other issues:
Conclusion is not finished and has some working comment (did the authors read the paper before submitting it at all?).
Conclusion is now finished. That was my fault.
The paper is about wearable sensors, so the wearability parameter seems to be excessive.
The authors do agree this is could be excessive, however the team have worked with sensors that have not satisfied usability and caused issues around comfort. However as devices have gotten more mature, this is becoming less of an overall issue, however still remains important for operational and ambulatory contexts, such as defense, energy and construction
Type of electrode parameter only concerns specific sensors. What about optical wearable sensors? Maybe some others? The authors should either concentrate on wearable sensors with electrode-based information acquisition and indicate this in the title and abstract, or consider other types of connections.
This is a fair point, to try and resolve this point, we reviewed the literature and found references that suggest whilst other types of sensors exist, electrode based are currently the most prevalent, but we do acknowledge novel sensors such as Magnetocardiogram and Ballistocardiography do exist, but these likely will bring some interesting considerations for the future, but are not mainstream or developed enough for real world operations.
The title speaks about "safety critical scenarios". Do the authors really concentrate on such scenarious? It doesn't look so.
Again a fair point, some of the literature is from safety critical domains, some of it isn't. Whilst the paper is intended to be utilised in safety critical domains, some of the results are quite generic. We acknowledge there does need further review into specific safety critical domains, however there is still a lack of overall guidance, that brings together a variety of literature.
Also, there is no methodology presented how to select sensors (as stated in the title) based on the presented assessment framework.
To resolve this point, the team developed a checklist and scoring system, that standardises scoring out of 100. It is simple and not overly complex, with the intention both academics and practitioners could use it as a tool to benchmark sensors against the framework and identify the best sensors for their needs.
Round 2
Reviewer 1 Report
Comments and Suggestions for Authors
The comments are as follows,
1. How to select papers to be surveyed is not necessarily introduced in detail, Fig. 1 could be deleted.
Comments on the Quality of English Languagenone
Author Response
We'd like to thank reviewer one again for their time and comments to help improve the manuscript. Your comments have really helped shape the paper and bring it up a level in regards to it's value.
How to select papers to be surveyed is not necessarily introduced in detail, Fig. 1 could be deleted.
- We kept figure one, just as a team we felt some readers may prefer a visual aid, and as it's not overly complex it helps add clarity to the paper selection process.
- We rewrote and added in some additional references to help add clarity to the reasoning behind the paper selection and why it solely focused on parameters directly related to the sensors.
Reviewer 2 Report
Comments and Suggestions for Authors
The authors have done a good job improving the paper. I accept the answers to the weak points identified in the previous review as well as the modeifications introduced.
There are some minor concerns left:
"must be improved":
- please, check the affiliation formatting: there should be no words “Affiliation 1” or “Affiliation 2”
- many of titles and subtitles have different capitalization – only the first letter (e.g., up to 3.2.4), all words capitalized (e.g., 3.2.5, 3.2.6, 3.3.4, 3.3.6.2), mixed (e.g., 3.3.0.1) or no capitalization at all (e.g., 3.3.6.1). Please, check all of them carefully.
- is the "0" eligible for section numbering (at least, 3.3.0.1 and 4.0)?
"suggestions, but the final decision is up to the authors":
- would be interesting to see some more information about the example case study (e.g., the scores givent by the respondents to the parameters), maybe and example of one or two sensors scored (could be placed into the same table).
- colors in the scoring table make it difficult to read.
- scoring table could be moved into the text (sec. 4) - it's just a less-than-one-page table.
Author Response
- We would like to the reviewer 2 for the comments and feedback, as well as their time helping us make the manuscript the best paper possible.
The authors have done a good job improving the paper. I accept the answers to the weak points identified in the previous review as well as the modeifications introduced.
There are some minor concerns left:
"must be improved":
- please, check the affiliation formatting: there should be no words “Affiliation 1” or “Affiliation 2”
- many of titles and subtitles have different capitalization – only the first letter (e.g., up to 3.2.4), all words capitalized (e.g., 3.2.5, 3.2.6, 3.3.4, 3.3.6.2), mixed (e.g., 3.3.0.1) or no capitalization at all (e.g., 3.3.6.1). Please, check all of them carefully.
is the "0" eligible for section numbering (at least, 3.3.0.1 and 4.0)?
- We have made all of these changes.
"suggestions, but the final decision is up to the authors":
- would be interesting to see some more information about the example case study (e.g., the scores givent by the respondents to the parameters), maybe and example of one or two sensors scored (could be placed into the same table).
- colors in the scoring table make it difficult to read.
scoring table could be moved into the text (sec. 4) - it's just a less-than-one-page table.
- Thank you for the kind and constructive comments about the checklist. I have removed all colouring bar a few rows, just to help improve visual clarity, hopefully now the checklist is much more readable.
- For better comparison and scoring, we changed the binary nature to a Likert scale, but also include a 0 score option if certain parameters are not relevant/available in some scorings. We have included two checklists, one for a PPG and the prior ECG. I also added some clarity that the data is just random data for the purpose of the paper.
- Rather than include the checklist in the paper itself, we included a Spider chart that allows for quick visual comparison between the two devices, with it being clear the ECG performs better.
- In the appendices we also included the formula to normalise scoring with instructions so it's clear and accessible.